# How Does Dietary Intake Relate to Dispositional Optimism and Health-Related Quality of Life in Germline *BRCA1/2* Mutation Carriers?

**DOI:** 10.3390/nu15061396

**Published:** 2023-03-14

**Authors:** Anne Esser, Leonie Neirich, Sabine Grill, Stephan C. Bischoff, Martin Halle, Michael Siniatchkin, Maryam Yahiaoui-Doktor, Marion Kiechle, Jacqueline Lammert

**Affiliations:** 1Department of Gynecology and Center for Hereditary Breast and Ovarian Cancer, University Hospital Rechts der Isar, Technical University of Munich (TUM), 81675 Munich, Germany; 2Institute of Nutritional Medicine, University of Hohenheim, 70599 Stuttgart, Germany; 3Department of Prevention and Sports Medicine, University Hospital Klinikum Rechts der Isar, Technical University of Munich (TUM), 81675 Munich, Germany; 4Clinic for Child and Adolescent Psychiatry and Psychotherapy, Medical Center Bethel, University of Bielefeld, 33617 Bielefeld, Germany; 5Institute for Medical Informatics, Statistics and Epidemiology, University of Leipzig, 04107 Leipzig, Germany

**Keywords:** *BRCA1*, *BRCA2*, DII, Mediterranean diet, HRQoL, metabolic syndrome

## Abstract

*Background*: The Mediterranean diet (MD) is an anti-inflammatory diet linked to improved health-related quality of life (HRQoL). Germline (g)*BRCA1/2* mutation carriers have an increased risk of developing breast cancer and are often exposed to severe cancer treatments, thus the improvement of HRQoL is important. Little is known about the associations between dietary intake and HRQoL in this population. *Methods*: We included 312 g*BRCA1/2* mutation carriers from an ongoing prospective randomized controlled lifestyle intervention trial. Baseline data from the EPIC food frequency questionnaire was used to calculate the dietary inflammatory index (DII), and adherence to MD was captured by the 14-item PREDIMED questionnaire. HRQoL was measured by the EORTC QLQ-C30 and LOT-R questionnaires. The presence of metabolic syndrome (MetS) was determined using anthropometric measurements, blood samples and vital parameters. Linear and logistic regression models were performed to assess the possible impact of diet and metabolic syndrome on HRQoL. *Results*: Women with a prior history of cancer (59.6%) reported lower DIIs than women without it (*p* = 0.011). A greater adherence to MD was associated with lower DII scores (*p* < 0.001) and reduced odds for metabolic syndrome (MetS) (*p* = 0.024). Women with a more optimistic outlook on life reported greater adherence to MD (*p* < 0.001), whereas a more pessimistic outlook on life increased the odds for MetS (OR = 1.15; *p* = 0.023). *Conclusions*: This is the first study in g*BRCA1/2* mutation carriers that has linked MD, DII, and MetS to HRQoL. The long-term clinical implications of these findings are yet to be determined.

## 1. Introduction

With continual improvements in cancer outcomes, both patients and clinicians are shifting their focus from survival alone towards improving health-related quality of life (HRQoL) and patient-centred functional outcomes [1]. HRQoL is defined as the impact a disease and its treatment have on a patient’s physical, functional, psychological, social, and financial well-being [2,3,4]. In cancer care, there is a growing recognition of the significance of HRQoL, as reduced HRQoL may result in lower treatment adherence [5] and an increased risk of mortality [6]. A more comprehensive definition of HRQoL could encompass dispositional optimism, which is a psychological attribute associated with health advantages [7]. Different aspects of HRQoL have been associated with chronic inflammation, i.e., decreased physical [8] and cognitive functioning [9], increased fatigue [10] and higher pain levels [11]. Pre-treatment inflammatory status may predict the development of common cancer treatment side effects [12], e.g., aromatase inhibitor-induced musculoskeletal syndrome in women with pre-existing musculoskeletal pain. Most importantly, elevated inflammatory markers have been associated with adverse cancer outcomes [13,14], potentially by promoting a microenvironment for tumour growth and metastasis [15]. The quantity, quality, and composition of foods have been shown to regulate inflammation [16,17,18]. This has prompted research into developing a literature-derived index to reflect the inflammatory potential of diets; the Dietary Inflammatory Index (DII) [19] scores an individual’s diet on a continuum from anti- to pro-inflammatory. A pro-inflammatory diet has been linked to an increased cardiovascular risk and mortality [20], and it increases the likelihood of both metabolic syndrome (MetS) [21] and various types of cancer [22,23,24,25]. Recent studies indicate a negative association between a pro-inflammatory diet and HRQoL [26,27,28]. A diet associated with low DII scores is the Mediterranean diet (MD) [29]. MD is characterized by high consumption of fruits, vegetables, legumes, grains, and polyunsaturated fats from olive oil and nuts, moderate consumption of fish and dairy products, and low intake of red meat and processed foods [30]. MD has been shown to be associated with reduced cardiovascular risk [31], prevent MetS and lower cancer risk [32]. Furthermore, adherence to MD has been linked to improved HRQoL in healthy individuals [33,34], as well as cancer survivors [35].

Breast cancer (BC) is the most common type of cancer in women [36]. Particularly vulnerable are women with a germline (g)*BRCA1/2* mutation have a risk of 69–72% of developing breast cancer and a risk of 17–44% of developing ovarian cancer by the age of 80 years [37]. These women are exposed to cancer treatments and/or prophylactic surgeries with detrimental short- and long-term effects on their health [38,39,40,41,42] and HRQoL [6,43,44,45]. Recent studies suggest that beneficial dietary changes after completing primary cancer treatment, as opposed to during treatment, might be most effective in improving HRQoL [46]. Dietary factors to reduce chronic inflammation and improve metabolic profile may be an approach to improving HRQoL, functional capacity, and cancer outcomes in women with a g*BRCA1/2* mutation. A first step in addressing this issue is to determine the relationship of DII, MD, MetS, and different aspects of HRQoL in g*BRCA1/2* mutation carriers with and without a previous history of cancer.

## 2. Methods

### 2.1. Study Design and Participants

The present study is a cross-sectional secondary analysis of the baseline data from the randomized controlled LIBRE-2 trial (a lifestyle intervention study in women with hereditary breast and ovarian cancer) and the associated feasibility study LIBRE-1 [47,48]. The trials are registered at ClinicalTrial.gov (NCT numbers: NCT02087592–registered on 14 March 2014, NCT02516540–registered on 6 August 2015). The LIBRE-2 trial is an ongoing, two-armed randomized (1:1) controlled multicentre trial conducted in Germany aimed at determining the impact of a structured one-year lifestyle intervention program on adherence to MD, cardiorespiratory fitness, and body mass index (BMI) among *gBRCA1/2* mutation carriers. The study cohort includes both women with a previous diagnosis of early stage cancer in remission (diseased) and without a prior cancer diagnosis (non-diseased). Details on the study design have been published elsewhere [47,48]. A total of 312 participants were available for the current analysis.

### 2.2. Instruments

*Blood samples, anthropometric measurements, and medical history.* At baseline, participants completed a standardized questionnaire to collect information on their medical history, socio-demographic factors, as well as lifestyle factors. Furthermore, all participants underwent a physical examination to determine systolic and diastolic blood pressure, heart rate, and anthropometric measurements such as height (in m), body weight (in kg), and waist and hip circumferences (in cm). These were used to calculate BMI (kg/m^2^) and the waist-to-hip ratio (waist circumference in cm/hip circumference in cm). Blood samples were taken after a 12-h fasting period, and analysed by the affiliated laboratories of the local institutions. MetS was defined according to the International Diabetes Federation criteria by the presence of a waist circumference ≥ 80 cm and at least two metabolic abnormalities, i.e., fasting glucose ≥ 100 mg/dL, systolic blood pressure ≥ 130 mmHg and/or diastolic blood pressure ≥ 85 mmHg, triglycerides ≥ 150 mg/dL, HDL-cholesterol < 50 mg/dL and/or treatment with lipid-lowering, glucose-lowering or antihypertensive drugs. Cardiopulmonary exercise testing was conducted to assess cardiorespiratory fitness via peak oxygen uptake (VO_2peak_).

*FFQ, MEDAS and Dietary Inflammatory Index.* Dietary intake was determined by two validated questionnaires. The participants completed the German version of the PREDIMED questionnaire, the Mediterranean diet adherence screener (MEDAS), a 14-item questionnaire that captures adherence to MD [49,50,51]. We calculated the MEDAS score as the percentage of positively answered questions [52]. Additionally, the German version of the EPIC food frequency questionnaire (FFQ) was applied to collect information on the quantity and frequency of 148 food items consumed over the previous year [53,54]. Data from the FFQ were then used to calculate DII using the method reported by Shivappa et al. [19]. Briefly, the DII is based on 1943 scientific papers scoring 45 food parameters according to whether they increased (+1), decreased (−1), or had no effect (0) on six inflammatory biomarkers (IL-1β, IL-4, IL-6, IL-10, TNF-α, and CRPs). As reported in previous studies [22,55,56,57], not all required food items were assessed by the German FFQ. Hence, the DII was calculated using the corresponding 30 food parameters available from the FFQ used in our study. Those were carbohydrates, protein, saturated fat, polyunsaturated fatty acids (PUFA), monounsaturated fatty acids (MUFA), n-3-fatty-acids, n-6-fatty-acids, cholesterol, total fat, energy, fibre, alcohol, iron, magnesium, zinc, vitamin A, thiamin, vitamin B12, riboflavin, niacin, vitamin B6, folic acid, vitamin C, vitamin D, vitamin E, flavonones, anthocyanidins, flavan-3-ol, flavonols, and flavones.

*Psychological questionnaires.* All LIBRE trial participants completed several psychological questionnaires. To assess optimism and pessimism as a personality trait, the revised 10-item life orientation test (LOT-R) was applied [58]. The “optimism score” (LOTR-O) ranging from 0 (minimally optimistic) to 12 (maximally optimistic) was calculated as the sum of the three positively formulated items. The “pessimism score” (LOTR-P) was calculated accordingly. The EORTC QLQ-C30 (questionnaire for quality of life assessment in patients with cancer, Version 3.0) [3] was used to evaluate HRQoL. This questionnaire consists of 30 items and is designed for patients receiving cancer treatment regardless of cancer type and location. It measures five functional dimensions (physical, role, emotional, cognitive, and social), three symptom items (fatigue, nausea or vomiting, and pain), six single items (dyspnea, insomnia, appetite loss, constipation, diarrhea, and financial impact), and a global health status, which is the mean of two questions regarding overall health and overall quality of life. The BKAE (“Bewertung körperlicher Aktivität und Ernährung”, in English “evaluation of physical activity and nutrition”) is a questionnaire designed specifically for the LIBRE trials [47,48] to analyse attitudes and views on physical activity and dietary intake. It is based on the concept of “planned behaviour” by Fishbein & Ajzen (1975) [59] promoting that attitude, subjective norms and perceived behaviour control contribute to behavioural intention, which leads to actual behaviour. We only used the dietary information of the questionnaire for our analysis. The scores *BKAE-AT* (attitude towards healthy eating), *BKAE-SN* (subjective norms about healthy eating), *BKAE-PBC* (perceived behaviour control over healthy eating), *BKAE-IT* (intention to eat healthy in the future) and *BKAE-PB* (past behaviour with regard to healthy eating) range from 0 (minimum) to 100 (maximum). The physical activity part of the questionnaire has been evaluated previously; the strongest predictor for cardiopulmonary fitness was attitudes towards physical activity [60].

### 2.3. Statistical Analysis

SPSS Version 29.0.0 (IBM Corp., Armonk, NY, USA) was used to analyse data. Descriptive statistics are presented as mean ± standard deviation (SD) for continuous variables or as proportions for categorical variables. The distributions of continuous variables between diseased and non-diseased women were compared using Student’s *t*-test. The distributions of categorical variables were compared using the Chi-square test. Linear regression models were created to detect associations between dietary intake and HRQoL. EORTC-QC30 scores were evaluated in diseased women only since the questionnaire was validated for cancer patients. Logistic regression models were performed to estimate odds ratios (ORs) and their associated 95% confidence intervals (95% CI) between MetS, dietary intake and different aspects of HRQoL. Multivariate analyses were carried out to control for potential confounding variables. These analyses were adjusted for body composition (BMI), physical fitness (VO_2peak_), adherence to MD, and/or dietary inflammatory potential (DII). All *p* values were based on two-sided tests and were considered significant if *p* ≤ 0.05.

### 2.4. Ethics

The study was approved by the ethics committees of both the host institutions Technical University of Munich (Reference No. 5685/13), the University Hospital Cologne (Reference No. 13-053), the University Hospital Schleswig-Holstein in Kiel (Reference No. B-235/13), and the participating study centres. Written consent from all study participants was obtained. All methods were carried out in accordance with relevant guidelines and regulations.

## 3. Results

A total of 312 women with a g*BRCA1* and/or g*BRCA2* mutation were included in the study. Table 1 summarizes the selected participants’ characteristics by health status (diseased vs. non-diseased). The mean age of the entire study cohort was 43.5 years (SD ± 10.3 years). Of all the women, 59.6% had a previous diagnosis of cancer. Among these, breast cancer accounted for 88.7% and ovarian cancer for 7.0% of all cancer cases. Women with a history of breast cancer were older (46.5 years vs. 39.1 years, *p* < 0.001), more likely married (67% vs. 55%, *p* = 0.026), and less educated (high school diploma: 58% vs. 75%, *p* = 0.002). Diseased women had significantly lower hsCRP levels (1.7 vs. 3.3 mg/L, *p* = 0.045) and lower DII scores (−1.1 vs. −0.5, *p* = 0.011) compared to non-diseased women. Non-diseased mutation carriers had better physical fitness (17.3 vs. 16 mL/min/kg, *p* = 0.029), reported significantly higher quality of life (QL2 72.2 vs. 57.7, *p* = 0.041), role (RF 90.3 vs. 79.8, *p* < 0.001), cognitive (CF 82.5 vs. 72.9, *p* < 0.001) and social functioning (SF 85.2 vs. 72.0, *p* < 0.001), and experienced less pain (PA 15.7 vs. 25.6, *p* = 0.001), dyspnea (DY 9.6 vs. 16.1, *p* = 0.015), insomnia (SL 28.0 vs. 39.4, *p* = 0.003), and fewer financial difficulties (FI 4.3 vs. 18.5, *p* < 0.001). On the other hand, diseased mutation carriers reported stronger social norms about healthy eating (BKAE-SN 79.6 vs. 73.7, *p* = 0.008) and greater behavioural control over healthy eating (BKAE-PBC 86.9 vs. 84.2, *p* = 0.010). They also reported a more frequent consumption of healthy foods compared to women without a prior history of cancer (BKAE-PB 58.5 vs. 51.6; *p* = 0.008).

We then analysed associations between DII and various metabolic and lifestyle factors using linear regressions. The results are presented in Table 2. A lower DII score was significantly associated with higher adherence to MD (*p* < 0.001). Among diseased women, higher DII scores were associated with better role functioning (RF) (*p* = 0.032), cognitive functioning (CF) (*p* = 0.003), and social functioning (SF) (*p* = 0.012) as well as decreased fatigue (FA) (*p* = 0.046), dyspnea (DY) (*p* = 0.029) and appetite loss (AP) (*p* = 0.007).

Associations between adherence to MD (MEDAS) and various factors were carried out using linear regressions (see Table 3). Adherence to MD was associated with higher VO_2peak_ (*p* = 0.024), as well as lower DII scores (*p* = <0.001). Furthermore, adherence to MD was associated with dispositional optimism (*p* = 0.001).

We then carried out logistic regression models to estimate odds ratios (OR) and their associated 95% confidence intervals (95% CI) of having metabolic syndrome (MetS) by different dietary variables and different aspects of HRQoL (see Table 4). Higher adherence to MD (MEDAS ≥ 0.50) reduced odds for MetS (OR = 0.538, *p* = 0.024). Women with dispositional pessimism had increased odds for MetS (OR = 1.147, *p* = 0.023). Among diseased women, those who had poorer physical functioning (OR = 0.955, *p* < 0.001) or experienced more dyspnea (OR = 1.017, *p* = 0.012) had increased odds for MetS.

## 4. Discussion

The aim of this analysis was to evaluate the relationship between anti-inflammatory diet, metabolic syndrome (MetS), and different aspects of health-related quality of life (HRQoL) in g*BRCA1/2* mutation carriers.

g*BRCA1/2* mutation carriers have a very high lifetime risk of developing breast and/or ovarian cancers. The average age of cancer diagnosis is substantially younger than in the general population (37). *BRCA*-associated cancers exhibit pathological features suggestive of an aggressive phenotype [61,62,63], and therefore, most patients undergo chemotherapy with detrimental side effects. When diagnosed with ER-positive breast cancer, patients might benefit from an extended adjuvant endocrine therapy [64,65], especially premenopausal women [66]. However, adjuvant endocrine therapy impacts HRQoL negatively [67]. Thus, identifying modifiable lifestyle factors to improve HRQoL is of particular relevance to g*BRCA1/2* mutation carriers, possibly resulting in better treatment adherence and (cancer-free) survival. 

We observed that diseased women consumed a more anti-inflammatory diet compared to non-diseased women (DII −1.1 vs. −0.5, *p* = 0.011). Moreover, diseased participants perceived greater behavioural control over selecting healthier food options and were more likely to make healthier food choices than women without a history of cancer. This conforms to previous research that a breast cancer diagnosis can lead to beneficial dietary changes [68]. The German breast cancer guideline issued by the German Association of the Scientific Medical Societies (AWMF) and the German Agency for Quality in Medicine (AeZQ) acknowledges the importance of lifestyle factors, such as diet and physical activity, in the aftercare of breast cancer patients. However, it was not until 2017 that the guideline included this recommendation [69]. The guideline suggests adhering to the dietary guidelines set by the German Society for Nutrition (DGE), which emphasize the consumption of plant-based foods such as cereal, grains, fruits, and vegetables as the foundation of a healthy diet, with small portions of animal products such as dairy, eggs, meat, and fish [70]. Compared to the MD, the consumption of olive oil, fish, seafood, and red wine is less emphasized in the DGE guidelines. As four items of the MEDAS focus on these food groups, it is possible that the lack of emphasis on them in the DGE guidelines may explain why diseased women in our study did not report a higher adherence to the MD compared to non-diseased women.

Of interest is the significant inverse association between adherence to MD and DII, indicating that MD is an anti-inflammatory diet (*p* = <0.001). In a prospective study, Hodge et al. (2016) identified MD as an anti-inflammatory diet that significantly reduced the risk of lung cancer [29]. The PREDIMED trial was the first randomized controlled trial to support these findings in a group of postmenopausal females; adherence to MD reduced the risk of breast cancer by 68% (95% CI 0.13–0.79) [69].

Porciello et al. [35] reported that adherence to MD in breast cancer survivors was associated with better HRQoL, i.e., improved physical functioning, better sleep quality and lower pain. We were not able to show an association between adherence to MD and HRQoL among diseased g*BRCA1/2* mutation carriers in our univariate and multivariate analyses. In our analysis, the median time from cancer diagnosis to study enrolment was four years (range: 1–48 years). To be eligible for participation in the LIBRE study, women had to be physically fit and functional, and several criteria that could hinder participation in the intervention program had to be excluded at study entry. These criteria included ongoing chemotherapy and/or radiation therapy, metastatic tumor disease, Karnofsky index below 60%, and exercise capacity below 50 watts. Consequently, our study participants were likely much fitter and more functional than those in Porciello et al.’s study, where women had to be diagnosed with breast cancer within the previous twelve months and had a mean age that was ten years older than our study participants.

In our study, adherence to MD was positively associated with dispositional optimism (*p* ≤ 0.001). This finding is consistent with the results of a study by Ait-Hadad et al. [70], which investigated the relationship between dietary intake and dispositional optimism in a sample of over 32,000 participants. The authors reported a positive association between optimism and overall diet quality; high intake of fruits, vegetables, legumes, whole grains, seafood, and fats was positively associated with optimism, while high intake of meat and dairy products was negatively associated with optimism. Dispositional optimism is characterized as a general expectation or belief in positive outcomes in the future [71]. It has been associated with improved cardiovascular health and reduced all-cause as well as cause-specific mortality in large epidemiological studies [72,73]. Among breast cancer patients, optimism has been linked to psychological well-being and improved quality of life [74,75]. Boehm et al. found that dispositional optimism was associated with higher serum levels of antioxidants [76]. This association was partially influenced by dietary intake. Scheier and Carver [77] suggest that there are two underlying mechanisms linking optimism to health. Firstly, dispositional optimism facilitates the engagement in health promoting behaviours, i.e., diet and physical activity. Secondly, optimistic individuals better cope with adverse life events better than pessimistic individuals, which results in reduced stress levels and increased physiological wellbeing. Although dispositional optimism is considered to be relatively stable across one’s lifespan, some studies found that cognitive therapy can increase optimism levels [78,79]. Since dispositional optimism and MD are linked in g*BRCA1/2* mutation carriers, identifying further strategies to increase dispositional optimism might help to implement a healthy diet. 

Moreover, adherence to MD was associated with reduced odds for MetS (OR = 0.54, *p* = 0.024). An Italian randomized controlled trial found that an MD-based dietary intervention in g*BRCA1/2* mutation carriers improved adherence to MD and reduced components of MetS [80]. Recent studies suggest that MetS is associated with impaired HRQoL [81,82,83]. Cohen et al. [84] reported a positive association between pessimism and the prevalence of MetS in patients with coronary heart disease. In our analysis, we found a positive association between MetS and dispositional pessimism (OR = 1.15, *p* = 0.023). In univariate analyses, MetS was associated with poorer physical functioning (OR = 0.96, *p* = <0.001) and higher levels of dyspnea (OR = 1.02; *p* = 0.012) among diseased women. However, these associations diminished following adjustment for physical fitness (VO_2max_).

In women with a history of cancer, higher DII scores were associated with better role functioning (RF), cognitive functioning (CF), and social functioning (SF), as well as reduced fatigue (FA), reduced dyspnea (DY), and reduced appetite loss (AP). These findings were robust after adjustment for body composition (BMI), physical fitness (VO_2peak_), and adherence to MD. To rule out that different types of cancer treatment or time since diagnosis influenced these associations, we calculated further multivariate regression models (see Appendix A). Our results were similar (see Table A1, Table A2 and Table A3). This is surprising since it is contradictory to prior findings indicating that a more pro-inflammatory diet is associated with reduced HRQoL [28]. A possible explanation could be that diseased women with better HRQoL were not concerned about healthy eating. According to the theory of planned behaviour by Fishbein and Ajzen, behaviours are influenced by intentions, which are determined by three factors: attitudes, subjective norms, and perceived behavioural control [59]. To test our hypothesis, we adjusted our multivariate regression models for associations between DII and different dimensions of EORTC QLQ-C30 for the three core components of the Fishbein and Ajzen model, i.e., attitudes, subjective norms and perceived behavioural control. None of the three factors were associated with a pro-inflammatory diet nor did they influence the link between greater DII scores and reduced role, cognitive and social functioning (see Appendix A—Table A4). After inserting the variables attitudes, social norms and perceived behavioural control into the linear regression models for DII and fatigue, dyspnea and appetite loss, the models no longer reached significant levels (*p* = 0.059–0.143). Thus, in our analysis, the positive associations between pro-inflammatory diet patterns and HRQoL were likely not influenced by attitudes and beliefs towards healthy eating.

### Strengths and Limitations

The strengths of the current study include the comprehensive evaluation of predictors that could be linked to HRQoL in g*BRCA1/2* mutation carriers. After adjusting for body composition, physical fitness and eating patterns, the adjusted and unadjusted results did not differ significantly. Therefore, any additional confounding was likely to be small. This supports our hypothesis that dietary intake is linked to different aspects of HRQoL. Although our results provide an interesting direction for HRQoL research, this study had several limitations. Firstly, the nature of a cross-sectional secondary analysis cannot establish a cause-and-effect relationship. The prospective nature of the LIBRE trials will allow for evaluating the impact of dietary changes on HRQoL. Secondly, as the number and type of food components to compute DII vary between studies, our results can hardly be compared to other populations of g*BRCA1/2* mutation carriers. Our study cohort was not representative for the average German population regarding education, net income, marital status, and parity [85,86,87,88]. Considering that our study cohort consisted of health-conscious females [89], the results obtained in this analysis likely underestimate the true associations between a pro-inflammatory diet and HRQoL outcomes. Finally, our cohort was not sufficiently powered to conduct analyses stratified by the g*BRCA* mutation type.

## 5. Conclusions

We were able to show that adherence to MD is linked to a more anti-inflammatory diet, dispositional optimism, and lower MetS prevalence among g*BRCA1/2* mutations carriers. Further research is needed to determine the long-term clinical implications of these findings.

## Figures and Tables

**Table 1 nutrients-15-01396-t001:** Baseline characteristics for participants with and without history of cancer.

Characteristic	Diseased	Non-Diseased	*p*-Value
*n* (%)	186 (59.6%)	126 (40.4%)	
Socio-demographic Data			
Age, years, mean ± SD	46.5 ± 9.2	39.1 ± 10.4	**<0.001 ***
Married, *n* (%)	(125) 67 %	70 (55%)	**0.026 ***
Number of children, mean ± SD	1.3 ± 0.9	1.1 ± 1.1	0.110
Education, *n* (%)			
• General university entrance qualification	104 (58%)	95 (75%)	**0.002 ***
• University degree	82 (44%)	68 (54%)	0.074
Net income, EUR, mean ± SD, *n*	4043.8 ± 1982.3; *n* = 126	3851.4 ± 2258.8; *n* = 84	0.515
Anthropometric Data			
BMI, kg/m^2^, mean ± SD	25.6 ± 4.9	25 ± 5.5	0.301
Waist-to-hip ratio, mean ± SD	0.8 ± 0.1	0.8 ± 0.1	0.256
Metabolic Data			
Metabolic syndrome, *n* (%)	45 (24%)	26 (21%)	0.426
VO_2peak_, mL/min/kg, mean ± SD	16.0 ± 5.0	17.3 ± 4.7	**0.029 ***
Nutritional Data			
DII, mean ± SD	−1.1 ± 1.8	−0.5 ± 1.2	**0.011 ***
MEDAS, mean ± SD	47.9 ± 16.6	46.5 ± 15.3	0.478
Psychological Data			
LOTR-O, mean ± SD	4.2 ± 2.9	4.1 ± 3.0	0.792
LOTR-P, mean ± SD	4.3 ± 2.1	4.0 ± 2.4	0.267
Quality of life (QL2), mean ± SD	67.7 ± 19.1	72.2 ± 8.6	**0.041 ***
Physical Functioning (PF2), mean ± SD	88.8 ± 12.6	91.4 ± 11.0	0.054
Role Functioning (RF2), mean ± SD	79.8 ± 24.0	90.3 ± 18.5	**<0.001 ***
Emotional Functioning (EF), mean ± SD	61.7 ± 27.3	62.6 ± 24.2	0.770
Cognitive Functioning (CF), mean ± SD	72.9 ± 25.9	82.5 ± 19.2	**<0.001 ***
Social Functioning (SF), mean ± SD	72.0 ± 30.0	85.2 ± 23.0	**<0.001 ***
Fatigue (FA), mean ± SD	33.6 ± 26.1	28.9 ± 19.9	0.085
Nausea and vomiting (NV), mean ± SD	3.9 ± 9.6	6.3 ± 14.9	0.096
Pain (PA), mean ± SD	25.6 + 28.3	15.7 ± 21.6	**0.001 ***
Dyspnea (DY), mean ± SD	16.1 ± 24.6	9.6 ± 20.7	**0.015 ***
Insomnia (SL), mean ± SD	39.4 ± 35.7	28.0 ± 29.8	**0.003 ***
Appetite loss (AP), mean ± SD	6.1 ± 16.9	4.0 ± 10.9	0.223
Constipation (CO), mean ± SD	10.0 ± 22.9	8.5 ± 20.3	0.543
Diarrhea (DI), mean ± SD	6.6 ± 15.8	11.5 ± 21.6	**0.033 ***
Financial difficulties (FI), mean ± SD	18.5 ± 29.0	4.3 ± 15.3	**<0.001 ***
BKAE-AT, mean ± SD	79.8 ± 6.8	78.9 ± 7.2	0.308
BKAE-SN, mean ± SD	79.6 ± 15.8	73.7 ± 18.7	**0.008 ***
BKAE-PBC, mean ± SD	86.9 ± 7.8	84.2 ± 10.7	**0.010 ***
BKAE-IT, mean ± SD	78.2 ± 9.8	76.2 ± 11.3	0.108
BKAE-PB, mean ± SD	58.5 ± 20.0	51.6 ± 23.0	**0.008 ***

* Results are statistically significant at a *p*-value of ≤0.05 (in bold).

**Table 2 nutrients-15-01396-t002:** Associations between DII and metabolic, lifestyle and HRQoL factors using linear regression models.

Characteristic	Mean ± SD	Unadjusted Estimate ^a^ (95% CI)	Unadjusted *p*-Value ^a^	Adjusted Estimate ^b^ (95% CI)	Adjusted *p*-Value ^b^
MEDAS	47.3 ± 16.8	−2.340 (−3.579; −1.101)	**<0.001 ***	−2.266 (−3.520; −1.011)	**<0.001 ***
Role functioning (RF2) ^1^	79.8 ± 24.0	0.012 (0.001; 0.023)	**0.032 ***	0.014 (0.003; 0.025)	**0.010 ***
Cognitive functioning (CF) ^1^	72.9 ± 25.9	0.015 (0.005; 0.025)	**0.003 ***	0.016 (0.006; 0.026)	**0.002 ***
Social functioning (SF) ^1^	72.0 ± 30.0	0.011 (0.002; 0.020)	**0.012 ***	0.013 (0.004; 0.021)	**0.005 ***
Fatigue (FA) ^1^	33.6 ± 26.1	−0.010 (−0.020; 0.000)	**0.046 ***	−0.012 (−0.022; −0.002)	**0.017 ***
Pain (PA) ^1^	25.6 ± 28.3	−0.009 (−0.018; 0.000)	0.057	−0.011 (−0.021; −0.002)	**0.017 ***
Dyspnea (DY) ^1^	16.1 ± 24.6	−0.012 (−0.022; −0.001)	**0.029 ***	−0.016 (−0.027; −0.005)	**0.004 ***
Appetite loss (AP) ^1^	6.1 ± 16.9	−0.021 (−0.036; −0.006)	**0.007 ***	−0.021 (−0.036; −0.006)	**0.008 ***

* Results are statistically significant at a *p*-value of ≤0.05 (in bold); ^1^ only for ‘Diseased’; ^a^ univariate linear regression unadjusted for BMI, VO_2peak_ and MEDAS; ^b^ multivariate linear regression adjusted for BMI, VO_2peak_ and MEDAS.

**Table 3 nutrients-15-01396-t003:** Associations between MEDAS and metabolic, lifestyle, and HRQoL factors using linear regression models.

Characteristic	Mean ± SD	Unadjusted Estimate ^a^ (95% CI)	Unadjusted *p*-Value ^a^	Adjusted Estimate ^b^ (95% CI)	Adjusted *p*-Value ^b^
VO_2peak_, mL/min/kg	16.7 ± 4.9	0.005 (0.001; 0.009)	**0.014 ***	0.004 (0.000; 0.008)	0.053
DII	−0.9 ± 1.9	−0.018 (−0.028; −0.009)	**<0.001 ***	−0.017 (−0.027; −0.008)	**<0.001 ***
LOTR-O	4.2 ± 2.9	0.011 (0.004; 0.017)	**<0.001 ***	0.010 (0.004; 0.016)	**0.002 ***

* Results are statistically significant at a *p*-value of ≤0.05 (in bold); ^a^ univariate linear regression unadjusted for BMI, VO_2peak_ and DII; ^b^ multivariate linear regression adjusted for BMI, VO_2peak_ and DII.

**Table 4 nutrients-15-01396-t004:** Associations between MetS, lifestyle and HRQoL factors using logistic regression models.

Predictor	Unadjusted OR ^a^ (95% CI)	Unadjusted *p*-Value ^a^	Adjusted OR ^b^ (95% CI)	Adjusted *p*-Value ^b^
High adherence to MD (MEDAS ≥ 0.50) vs. low adherence to MD (<0.50)	0.538 (0.314; 0.922)	**0.024 ***	0.602 (0.343; 1.058)	0.078
LOTR-P	1.147 (1.019; 1.292)	**0.023 ***	1.150 (1.012; 1.307)	**0.032 ***
Physical functioning (PF2) ^1^	0.955 (0.930; 0.980)	**<0.001 ***	0.963 (0.937; 0.990)	**0.007 ***
Dyspnea (DY) ^1^	1.017 (1.004; 1.030)	**0.012 ***	1.013 (0.999; 1.026)	0.061

* Results are statistically significant at a *p*-value of ≤0.05 (in bold); ^1^ only for ‘Diseased’; ^a^ univariate logistic regression unadjusted for VO_2peak_ and MEDAS; ^b^ multivariate logistic regression adjusted for VO_2peak_ and MEDAS.

## Data Availability

Data are available upon reasonable request to the corresponding author.

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
