# Peer review of "How Does Dietary Intake Relate to Dispositional Optimism and Health-Related Quality of Life in Germline BRCA1/2 Mutation Carriers?"

_nutrients, 2023, doi:10.3390/nu15061396_

Round 1

Reviewer 1 Report

The authors of this manuscript aimed to investigate a possible association between an anti-inflammatory diet, the metabolic syndrome, and various aspects of health-related quality of life in women who carry the gBRCA1/2 mutation, a mutation that makes them susceptible to developing breast and/or ovarian cancer. To do so, they analyzed baseline data from two existing studies, the randomized controlled "Lifestyle intervention study in women with hereditary breast and ovarian cancer" (LIBRE-2) and the related feasibility study LIBRE-1.

The manuscript is well written, the data are well presented, and the conclusions drawn are understandable and reasonable.

Author Response

Dear Sir or Madam,

thank you for reviewing our manuscript! The other reviewer had some questions and comments regarding the manuscript. Thus, we revised our manuscript. Please see the attachment for the revised manuscript and the cover letter providing point-by-point responses to the comments of the other reviewer.

Again, thank you very much for reviewing our manuscript and for your positive feedback!

Yours sincerely,

Anne Esser

Reviewer 2 Report

This manuscript investigated the associations between the health-related quality of life and different factors among germline BRCA1/2 mutation carriers.  This study was well designed and conducted. The data presented in the study is interesting to readers, but there are still some concerns:

1. The title of the manuscript is not quite appropriate. The associations of dietary intake with the health-related quality of life are only part of the presented data. I would suggest modifying the title so that it reflects the entire data better. 

2. It is better to report significant results only in table 2-4. 

3. As indicated at line 240, “diseased women consumed a more anti-inflammatory diet compared to non-diseased women (DII -1.1 vs. -0.5, p = 0.011)”, but why didn’t the MEDAS, which is an anti-inflammatory diet, show any significant difference between these two groups?

4. As indicated at line 252-255, the results in this manuscript didn’t align with previous studies, why?

Author Response

Dear Sir or Madam,

we thank you for your helpful comments! We revised the manuscript accordingly. Please see the attachment for our point-by-point responses to your comments.

Thank you!

Yours sincerely,

Anne Esser
